# OpenReview forum: "Controllable Safety Alignment: Inference-Time Adaptation to Diverse Safety Requirements"
_ICLR.cc/2025/Conference — ICLR 2025 Poster_

### Official Review · Reviewer_6dSX · 2024-10-28

**Soundness:** 3
**Presentation:** 3
**Contribution:** 3
**Rating:** 8
**Confidence:** 3

**Summary:**

The current safety alignment of large language models (LLMs) lacks flexibility and requires re-training. To address this, the authors propose CoSAlign, a data-centric method for LLMs to adapt to different safety configurations. They also develop a controllability evaluation protocol that considers helpfulness and configured safety, summarizing them into CoSA-Score. They construct CoSApien, a human-authored benchmark based on real-world LLM use cases with diverse safety requirements. CoSAlign leads to substantial gains in controllability over strong baselines, encouraging better representation and adaptation to pluralistic human values, increasing LLMs' practicality.

**Strengths:**

+ The proposed CoSAlign method offers flexibility by allowing models to be adapted to diverse safety requirements without the need for retraining.
The introduction of a new controllability evaluation protocol that balances both helpfulness and safety, summarized into a single CoSA-Score, is novel.

**Weaknesses:**

- Some parts are not easy to follow.
- The core contribution of the work is not clear. It is more like multiple small points mixed up.
- While the method offers flexibility with safety configs, it may face challenges in generalizing across highly diverse or conflicting safety requirements.

**Questions:**

1. Section 3: The evaluation protocol is like the one shown in "Lin, Stephanie, Jacob Hilton, and Owain Evans. "Truthfulqa: Measuring how models mimic human falsehoods." arXiv preprint arXiv:2109.07958 (2021)." What are the advantages of your protocol?

2. Section 5: The CoSAlign pipeline is more like a rule-based method. I doubt its generalization ability in real-world deployment. Moreover, this part is kind of messy and not easy to follow.

---

> ### Author Response · Authors · 2024-11-17
> **Response to reviewer 6dSX**
>
> We sincerely appreciate reviewer `6dSX` for their thoughtful review. We are glad that you find our method flexible and our evaluation protocol novel. We hope our response addresses your questions and concerns:
>
> ## Response to Weaknesses
>
> > The core contribution of the work is not clear. It is more like multiple small points mixed up.
>
> Summarized in Line 92-96, our core contributions are introducing the Controllable Safety Alignment framework and formulating the task of efficiently adapting models to diverse safety requirements at inference time. Our framework rethinks the current paradigm of safety alignment and enables LLMs to be responsibly deployed in diverse settings.
>
> To construct this comprehensive framework, we propose the CoSA-Score evaluation protocol, a human-curated benchmark CoSApien, and CoSAlign, a method for improving the controllability of LLMs. We believe all of these contributions are crucial for effectively adapting LLMs to diverse use cases. By providing this comprehensive set of artifacts, we enable rich future work on controllable safety alignment as pointed out in our discussion section.
>
>
>
> > While the method offers flexibility with safety configs, it may face challenges in generalizing across highly diverse or conflicting safety requirements.
>
> We have already conducted experiments in Section 6.1 (discussed in Line 450-464) to evaluate the impact of the distribution shift from simpler safety configs to more complex ones. We refer the reviewer to that section of a detailed discussion. We summarize it here again for your convenience: we train on CoSAlign-Train, which only contain synthetic categorical risk categories, and test on CoSApien, our proposed benchmarks consist of complex, fine-grained real-world configs (see examples in Appendix A.12). Human evaluation (Table 4) shows that CoSAlign still possesses strong controllability in safety configs that are more complex than those the model was trained on, and outperforms all baselines. This shows CoSAlign’s strong generalization ability from simpler training configs to more complex test configs.
>
>
>
> Finally, if there is a specific section you found unclear or uneasy to follow, please don’t hesitate to let us know!
>
> ## Response to questions
>
> > Section 3: The evaluation protocol is like the one shown in "Lin, Stephanie, Jacob Hilton, and Owain Evans. "Truthfulqa: Measuring how models mimic human falsehoods." arXiv preprint arXiv:2109.07958 (2021)." What are the advantages of your protocol?
>
> We’d like to clarify that while both our protocol and TruthfulQA use LLM as a judge, CoSA-Score use two separate judge-help and judge-safe to measure helpfulness and configured safety, and summarizing them into a single score that take account of both aspects. We argue this is a novel way to depict the tradeoff between helpfulness and safety aspects that allows distinguishing nuances such as a response that is both unsafe and helpful should be penalized more than a response that is unsafe but not helpful (for example, if the prompt requests criminal advice, a highly helpful response would likely cause greater harm to society than one that is less helpful).
>
>
>
> > Section 5: The CoSAlign pipeline is more like a rule-based method. I doubt its generalization ability in real-world deployment. Moreover, this part is kind of messy and not easy to follow.
>
> We’d like to clarify that because CoSAlign selects response pairs through the error score derived from LLM judge outcomes, it is not a rule-based method. On the other hand, a recent work on rule-based rewards has recently shown to be effective for controlling nuanced aspects of safety [1], so we agree there is indeed room to explore in the rule-based space.
>
> Related to CoSAlign’s generalization ability in real-world development, we have created the human-authored benchmark, CoSApien, with the help of professional red teaming specialists who are adept at real-world safety scenarios (detailed in Section 4). Thus we believe that CoSApien covers common safety scenarios in practice. In Table 4, Section 5.2, human evaluation shows that CoSAlign not only maintains high controllability but also outperforms all baselines, demonstrating the real-world applicability of our method. Due to the space constraints, we had to move some details of CoSAlign to the appendix, but we will continually iterate to improve the clarify in this part.
>
>
>
> We also conducted additional experiments on adversarial robustness, over-refusal, and combining CoSAlign with Cascade methods. Please find more details in the **“Response to all reviewers”** section. Thank you!
>
>
> ## References
> [1] [Rule Based Rewards for Language Model Safety](https://cdn.openai.com/rule-based-rewards-for-language-model-safety.pdf)

---

> > ### Comment · Reviewer_6dSX · 2024-11-18
> >
> > Thanks for the detailed comment! I would like to raise my score.

---

### Official Review · Reviewer_c6ps · 2024-11-03

**Soundness:** 2
**Presentation:** 3
**Contribution:** 3
**Rating:** 8
**Confidence:** 3

**Summary:**

The paper tackles the problem of LLM alignment through the use of safety configs that can be modified at inference time, allowing for more controllable and diverse alignment, while maintaining helpfulness. In the proposed method, LLMs are finetuned not by rewarding the response of the model with respect to the prompt, but also with respect to diverse safety configs, in order to generalize to unseen safety configs which have different safety requirements. The authors conduct a thorough evaluation of their relative method to existing ones.

The proposed method helps remove the ambiguity of alignment and helpfulness definitions from previous works by scoring model responses not only with respect to queries but also with respect to safety configs.

**Strengths:**

- Topic of interest - specialized safety configs is a topic of interest for practical use of LLMs in organizations that have different safety requirements.
- Alignment that properly incorporates both safety and helpfulness is important. Specifically the dataset used that couples safety and helpfulness is interesting.
- Finetuning a model on conversations conditioned on diverse policies for alignment seems to be a good strategy to overcome the genericness of safety responses.
- Most interesting in my opinion is that the concept of adding a safety config as a prefix for a conversation removes ambiguity from the definition of safe and helpful - instead of relying on an existing dataset that teaches a specific version of safe and helpful, you teach a model the concept of safe and helpful with respect to a safety config, which is more powerful and less ambiguous (it is "easy" to judge if a response is safe and helpful with respect to a safety config, it is hard and not well-defined without one).

**Weaknesses:**

- Use of the CoSA-score for evaluation - something a bit concerning to me about the CoSA-score is the way it merges safety and helpfulness into one value. For example, a negative score can rise without safety being altered at all simply by making the model less helpful (if helpfulness drops to zero, the score becomes zero which is better than negative). Similar things can also happen with positive scores and this indeed does seem to happen in some of the reported results, where the score rises mainly by reducing helpfulness (table 3, "seen configs", "in-context alignment", INST+ICA+5shot vs SFT+ICA+5shot). Furthermore, since the helpfulness scores are between {0,1,2,3,4,5}, you may lose a lot of information - e.g. if 100% of responses are safe, you get the same score in the case that they all have helpfulness = 3 and in the case that half are 0 and half are 5, which are somewhat different cases, as in the first the model is essentially helpful on all prompts and in the second only on half. Thus the other numbers reported in the tables are important as they do remove this ambiguity - by reporting helpful+safe and helpful+unsafe you get a complete picture. I did not see a discussion on issues such as these that may arise from the CoSA score, I believe it needs to be discussed.
- Higher helpfulness and lower safety (relative to cascade methods) - following the above point, when separating helpful+safe and helpful+unsafe, indeed CoSAlign has a higher helpful+safe score, but also a higher helpul+unsafe score than the cascade methods. I am not sure how to compare CoSAlign's 43% safe+helpful and 8% unsafe+helpful with the SFT+Cascade+Oracle's 29% safe+helpful and 0% unsafe+helpful. While having a lower safe+helpful score, Cascade seems very robust to being unsafe compared to CoSAlign, which is not reflected in the CoSA-score, and I think needs to be discussed.
- Related works - while the main selling point of the method proposed in this work is safety diversity, it also has a lot to do with the topic of maintaining the trained model both safe and helpful, I believe this can be discussed and more related works need to be mentioned in such a discussion (to give a few examples [1,2,3]).
- All that being said, I do appreciate the approach and am open to a fruitful discussion.


[1] - https://arxiv.org/abs/2309.07875

[2] - https://arxiv.org/abs/2308.01263

[3] - https://arxiv.org/abs/2401.16332

**Questions:**

- Is there some way to combine the cascades method with the CoSAlign method?

---

> ### Author Response · Authors · 2024-11-17
> **Response to reviewer c6ps**
>
> We deeply appreciate reviewer `c6ps` for their thoughtful feedback. We are thankful for your recognition of the practical relevance of our topic, the advantages of our proposed alignment method, and most interestingly, the framework of obtaining controllability through safety configs. We now address your questions and concerns below.
>
>
>
> ## Response to weaknesses
>
> > For example, a negative score can rise without safety being altered at all simply by making the model less helpful (if helpfulness drops to zero, the score becomes zero which is better than negative). Similar things can also happen with positive scores
>
> Thanks for highlighting this point! We argue that instead of a weakness, this is characteristic actually a benefit of CoSA-Score: We believe that unsafe responses that are *less helpful* should indeed be preferred over unsafe responses that are *more helpful*. For instance, if the prompt requests criminal advice, a highly helpful response would likely cause greater harm to society than one that is less helpful. By reducing helpfulness in unsafe responses, the potential negative impact is mitigated, aligning with our goal of promoting safer model behavior. Importantly, it is only by considering safety and helpfulness together, as CoSA-Score does, that such nuanced trade-offs can be effectively addressed. For positive scores, CoSA-Score's construction allows safe responses that are more helpful to be preferred over safe responses that are less helpful, as one would intuitively expect. We’d love to provide further clarifications and engage in further discussions on this design if you are interested.
>
>
>
> > Furthermore, since the helpfulness scores are between {0,1,2,3,4,5}, you may lose a lot of information [...] Thus the other numbers reported in the tables are important as they do remove this ambiguity - by reporting helpful+safe and helpful+unsafe you get a complete picture.
>
> We agree on your point that there might be more than one situation where the aggregated CoSA-Score is the same (e.g., all responses are somewhat helpful v.s. half responses are very helpful & the other half is not helpful at all). This is the tradeoff we are making by summarizing safety and helpfulness into a single score. We believe this design is still very meaningful because it allows us to (1) obtain an aggregated metric across all safety configs and prompt distributions to measure overall controllability (2) take into consideration of the nuanced trade-offs between helpfulness and safety together (related to the first point above). As you mentioned, we also report the helpful+safe and helpful+unsafe responses as focused metrics on positive-scored and negative-scored responses, making it easier to capture the complete picture. Thanks so much for pointing out these important nuances and we will add corresponding discussions to our draft.
>
>
>
> > Higher helpfulness and lower safety (relative to cascade methods)
>
> We acknowledge that CoSAlign has a drastically higher rate of helpful+unsafe responses at the cost of slightly higher rate of helpful+unsafe responses relative to the Cascade methods. Note that this result is not so surprising, given that Cascade methods specifically focus on improving safety by filtering out unsafe responses deemed by a classifier and replacing them with refusals. Therefore, in this case, the balance between helpfulness and safety is very important because a very safe but not helpful response is useless. As determined by the much higher CoSA-Score, which is specifically designed to capture the trade off between helpfulness and safety, CoSAlign clearly outperforms Cascade method. Also note that Cascade-Oracle is a very strong baseline that uses the evaluator model to conduct filtering, since it’s guaranteed to have 0% unsafe responses. Even in this case, CoSAlign outperforms Cascade-Oracle because there is a limited rate of helpful+safe responses.
>
> Inspired by your suggestions, we have also applied the Cascade methods on top of CoSAlign methods and Cascade-Oracle demonstrated additional gains. Please see **“Response to all reviewers”** section for details.
>
> **On related works**: Thank you so much for your suggestion. We agree that these can be a nice addition to our related work discussion. We will add a dedicated paragraph on balancing safety and helpfulness.

---

> ### Author Response · Authors · 2024-11-17
> **Response to reviewer c6ps - continued**
>
> ## Response to questions
>
> > Is there some way to combine the cascades method with the CoSAlign method?
>
> Yes! The Cascade method can be naturally incorporated into any model because it adds a post-hoc filtering step, which replaces unsafe responses deemed by the filtering model with refusals. This is similar to what deployed systems usually do (e.g., OpenAI’s content filter on ChatGPT). Although the filtering model is usually smaller than the generator model, we use (1) the generator model itself (2) the evaluator model as filtering model to construct this strong baseline. We have included results of combining Cascade with CoSAlign in the **“Response to all reviewers”** section.

---

> ### Author Response · Authors · 2024-11-23
> **Invitation for Further Discussion**
>
> Dear Reviewer `c6ps`,
>
> We sincerely appreciate your thoughtful feedback and have provided clarifications to your questions, including conducting new experiments on combining CoSAlign with Cascade methods as per your request. Your insights have been very constructive, and we would greatly value your review of our responses to ensure we have fully addressed your concerns.
>
> Please don’t hesitate to share any additional comments and we are eager to engage with you further before the discussion period ends. Thank you for your time and consideration.

---

> > ### Comment · Reviewer_c6ps · 2024-11-25
> >
> > I would like to thank the authors for their detailed responses and especially for showing that their CoSAlign method can be incorporated with the cascade methods. The results on the seen configs of the dataset look promising - as one might hope, it appears to obtain "the best of both worlds". One final follow-up question that I have in order to solidify this point is regarding the unseen split of the dataset that the authors say "follows the same pattern". What are the exact results?

---

> > > ### Author Response · Authors · 2024-11-25
> > > **Response to follow-up question**
> > >
> > > We sincerely appreciate reviewer `c6ps` for your recognition of our response and additional results. In the “Response to all reviewers” section we included the results on the seen split for conciseness, and now we present the exact results on combining CoSAlign with Cascade methods on the unseen split:
> > >
> > >
> > >
> > > | Model                                | CoSA-Score | Help+safe | Help+unsafe |
> > > |--------------------------------------|------------|-----------|-------------|
> > > | Llama3.1-8B-Instruct                 | 0.091      | 14.7%     | 2.9%        |
> > > | Llama3.1-8B-Instruct+Cascade         | 0.095      | 13.4%     | 1.5%        |
> > > | Llama3.1-8B-Instruct+Cascade-Oracle  | 0.119      | 14.7%     | **0.0%**    |
> > > | L3.1-8B-Inst-CoSAlign                | 0.293      | 42.8%     | 8.0%        |
> > > | L3.1-8B-Inst-CoSAlign+Cascade        | 0.274      | 36.6%     | 4.0%        |
> > > | L3.1-8B-Inst-CoSAlign+Cascade-Oracle | **0.364**  | **42.8%** | **0.0%**    |
> > >
> > >
> > >
> > > As we see above, similar to the results on the seen split, applying Cascade on CoSAlign-tuned model can effectively reduce the rate of helpful+unsafe responses, at the cost of slightly decreased helpful+safe responses, trading-off helpfulness for increased safety. The combination of CoSAlign and Cascade-Oracle continues to achieve the highest CoSA-Score, helpful+safe responses, and the lowest helpful+unsafe responses across all methods, demonstrating the "best of both world” when two methods are fused together.
> > >
> > >
> > > Thanks again for your response. Please let us know if you have any additional questions or comments, and we are happy to continue this engaging and productive discussion!

---

> > > > ### Author Response · Authors · 2024-11-29
> > > > **Further discussion**
> > > >
> > > > Thanks again reviewer c6ps. We would love to hear back from you regarding our previous response. Have we addressed your question adequately? We are happy to engage in further discussion in the remaining few days.

---

> > > > > ### Comment · Reviewer_c6ps · 2024-11-29
> > > > >
> > > > > Thank you for providing the results for these experiments.
> > > > > My initial worry was that CoSAlign improves helpfulness but compromises safety too much, as trading the relatively robust safety of cascade in favor of helpfulness feels like a potentially problematic tradeoff.
> > > > > However, the combined method of CoSAlign+cascade does appear to greatly improve helpfulness without compromising safety too much compared with other methods.
> > > > > With this my concerns are addressed, and after going over the other reviews and responses, I have decided to raise my score.

---

### Official Review · Reviewer_pCoa · 2024-11-04

**Soundness:** 2
**Presentation:** 3
**Contribution:** 2
**Rating:** 6
**Confidence:** 3

**Summary:**

This paper addresses the problem of having custom safety configurations (configs) for diverse uses of LLMs. The primary method uses preference-learning to align the LLMs such that they can follow the custom safety configs provided in their system prompt during inference. The results show increased safety controllability over existing approaches, suggesting the efficacy of the method.

**Strengths:**

1. The paper presents an interesting instance of preference learning applied to have customized safety, which may be of practical importance for effective and safe usage of LLMs in diverse applications.
2. The work contributes a new human-designed benchmark, CoSApien, for controllable safety alignment.
3. The claims of the paper are well-supported by human evaluations to show correctness of the assessments of the methods.

**Weaknesses:**

1. It may not be feasible to specify all aspects of safety, even by domain experts, in natural language. It's tedious and requires a lot of manual effort. Hence, the authors should consider and report the practical manual overhead of their alignment method, compared to the traditional methods where there are universal notions of safety to some general extent and it does not need to be redefined everytime (especially the commonly desired configs).
2. I don't see how the safety setting can be significantly different from the non-safety setting studied in other similar works on plurality alignment, necessitating the methods of the paper.
3. To determine "disallowed" content, we would need some kind of a bigger set of safety guidelines, from which those corresponding to the specified safety configs are removed and the remaining are disallowed. There is no mention of such a set of safety guidelines. Specifically, how would one systematically design prompts that elicit disallowed content, such as that in line 180?
4. I think it may be advantageous to evaluate CoSApien also with the kinds of content that an actual LLM can generate for the different kinds of prompts, for a real proof of its prompts being effective.
7. There is a lack of details about how the risk taxonomy is constructed. Line 295 says it is made from training data, but doesn't mention how. The footnote on page 6 says that the taxonomy consists of fewer categories and shorter definitions, which suggests that it is human-made.
8. The experiments are limited to Llama-3.1 8B and GPT-4o models. It would be interesting to see how CoSAlign would work with other LLMs such as Mistral.
9. I would suggest that the authors add some justification for not doing DPO on GPT-4o around line 472.
10. The authors should discuss the generally lower (hence better) helpful+unsafe values of the Cascade method over CoSAlign.
12. Terminology used before definition:
    1. Line 87: "training prompts"
    2. Line 88: "error-scoring mechanism". What is *error* here?
    3. Line 235: although the section for CoSAlign-Test is given, I think it should be described before using.
    4. Line 343: "data generator model"? Is it different from a language model?
    5. Line 484: "*overgeneralization* to disallowed content"?
13. Typos:
    1. Line 146: "provide" -> "provider"
    2. Line 163: "non-authorized" -> "authorized"
    3. Table 1 caption: "deteriorates" -> "deteriorated"
    4. Line 372: "controllability" -> "controllable"
    5. Table 3 has repeated setups 1 and 3 under CoSAlign methods.
    6. The legend for Figure 5 says that the red bars are for ICA, but the text says that it is for SFT. I think all of ICA, SFT, and CoSAlign should be shown in that figure.

**Questions:**

1. What would happen for ambiguous or contradictory safety configs? What if they contradict the system prompt that follows the safety configs during inference?
3. Line 114-115: "However, because of the complexity of
safety configs and the difficulty of constructing high-quality demonstrations at scale" - the complexity of safety configs is a problem for this work too. Moreover, in-context learning requires only a handful of demonstrations, hence the argument of absence of demonstrations *at scale* is void. Hence, I think these arguments against in-context learning are not very strong.
4. Line 183: How do you check/verify coverage?
5. How would GPT-4's own safety alignment affect its responses as judge-safe(.) in line 188?
6. Can the CoSA-Score be made more useful by assigning -1 to refusals/unhelpful responses?
7. Line 268: Are the training prompts corresponding to or independent of any safety configs used in the training?
8. Can the risk taxonomy made in the paper used in place of the other prior taxonomies, or is it specially tailored to CoSA?
9. Is $C_{i,j}\in R$, in line 313?
10. How is judge-help conceptually different from judge-addr?
11. Line 350: Why are allowed risks *penalized* by $\alpha$? I think they should be rewarded, as the model is adhering to the safety config.
12. Line 363: Why is the adversarial partition of WildGuardTrain removed from the training set?
13. Line 405: Manually checking that the test set contains all the 3 kinds of prompts is ok, but what is ensuring that to be the case in general for all the prompts? As far as I understand, it is just a dataset containing some prompts, not necessarily according to some safety configs.
14. Line 407: Does "helpful" mean that the judge-help outputs anything > 0 or equal to 1? Similarly, what values of the helpfulness scores from humans are considered to be "helpful" in Table 4?

---

> ### Author Response · Authors · 2024-11-17
> **Response to reviewer pCoa**
>
> We genuinely appreciate reviewer `pCoa` for their insightful review. We’re thankful that you recognize our work contributes to the “practical importance for effective and safe usage of LLMs in diverse applications,” and our paper is well-supported by human evaluations. We address your questions and concerns below:
>
>
>
> ## Response to weaknesses:
>
> **On “It may not be feasible to specify all aspects of safety, even by domain experts, in natural language”**: We’d like to clarify that safety configs do not necessarily need to specify “all aspects of safety”: just like what we do in our synthetic configs (exemplified in Appendix A.11), the safety config be used as providing exceptions to regular “one-size-fits-all" alignment, indicated by the instruction “On the other hand, you must continue to ensure safety in all other categories.” Therefore, only the difference between the desired safety behavior and a “one-size-fits-all” safety aligned model need to be specified, greatly reducing the overhead of producing safety configs.
>
> In practice, we show that starting CoSAlign from a one-size-fits-all safety aligned model can effectively retain safety on input distributions that are not directly specified in the safety config. In this case, the model will “default” to the one-size-fits-all safety behavior. In our new experiments shown in **“Response to all reviewers”** section, we find that CoSAlign can significantly increase adversarial robustness against popular jailbreak attacks and reduce over-refusal, while these behaviors are not directly specified in the safety config at all.
>
>
>
> **On the difference between our work and pluralistic alignment in non-safety settings**: Safety is a special setting because while current safety alignment techniques make the model robust to adversarial attacks, they also severely curtail the steerability of models on safety. Instructing models to modify safety standards can be seen as a jailbreak, thus not possible with one-size-fits-all safe models. We experimentally show this is indeed the case due to the ineffectiveness of in-context alignment in Section 5.1. On the other hand, for non-safety settings such as cultural alignment, techniques such as Anthropological Prompting [1] has shown to be an effective baseline. However, similar prompting-based techniques do not apply to the safety settings, motivating our CoSA framework and CoSAlign approach. Additionally, we conduct thorough safety-specific evaluation and propose metrics, benchmarks and data synthesis methods that are tailored to the safety-specific settings. All these contributions are novel and essential for studying pluralistic safety alignment.
>
>
>
> **On determining disallowed content**: Similar to your first point on “feasibility to specify all aspects of safety”, we’d like to clarify that since we start with a one-size-fits-all safety aligned model, the safety guideline only need to specify the difference between the “default” safety behavior. This design makes the model robust to prompts not specified in the config, as shown in results in the “Response to all reviewers” section.
>
> In general, determining what should be the disallowed content is a key sociotechnical problem and ongoing research. Approaches such as Constitutional AI [2], Collective Constitutional AI [3], and Simulated Deliberative Democracy [4] have been proposed to import societal value to determine what should be considered allowed/disallowed. We propose an alternative approach where this boundary can be flexibly and efficiently adapted during test time by adjusting the specification in safety config.
>
> >  I think it may be advantageous to evaluate CoSApien also with the kinds of content that an actual LLM can generate for the different kinds of prompts, for a real proof of its prompts being effective.
>
> We’d like to clarify that our CoSApien benchmark curated real-world safety configs by professional red teaming specialists who are adept with real-world safety scenarios. We also collected diverse human-written prompts for each config, which are natural and what actual LLMs are often asked about. Please refer to Section 4 for more details.
>
> **On details about risk taxonomy construction**: due to space constraints, we had to move some of the details to appendix, and please see a detailed description in Appendix A.2. To create this risk taxonomy, we first perform prompt clustering, and produce cluster descriptions with GPT-4o. Next, we manually edit the descriptions of largest clusters and produce the final risk taxonomy. We will also make the reference to the appendix clearer in the main text.

---

> ### Author Response · Authors · 2024-11-17
> **Response to reviewer pCoa - continued**
>
> **On “The experiments are limited to Llama-3.1 8B and GPT-4o models”**: We experiments on four model variants, Llama-3.1-8B-Instruct, Llama-3.1-8B-SFT, GPT-4o, and GPT-4o-mini, within the Llama and GPT model families, and covers both popular open-source and proprietary models. Thank you so much for your suggestions of more models. We believe that the Llama model family is a good representation of open-source models, thus we don’t see an urgent need to experiment with a similar open-source model Mistral, but we’ll take that into consideration.
>
>
>
> > I would suggest that the authors add some justification for not doing DPO on GPT-4o around line 472.
>
> Thanks for mentioning this point — we actually already have a footnote on page 9 that clarifies only SFT is publicly available for GPT.
>
>
>
> > The authors should discuss the generally lower (hence better) helpful+unsafe values of the Cascade method over CoSAlign.
>
> We have added additional results of CoSAlign+Cascade methods and included extended discussions. Please see the “Response to all reviewers” section for more details.
>
>
> Thank you so much for your comments on improving the terminology use & fixing typos. We will make sure to fix these issues in the manuscript.
>
>
> ## Response to questions
>
> **On ambiguous or contradictory safety configs**: While it’s challenging to formally define what configs are ambiguous and modeling ambiguity in general [5], qualitatively we find that when a CoSAlign-tuned model face prompts that are under-defined in config, or low-quality configs in general, the model will “default” to standard safety behavior as achieved by the one-size-fits-all aligned base model. We find this phenomenon very interesting because it shows that the model is protected by regular safety alignment when configs do not apply.
>
>
>
> > Line 114-115: "However, because of the complexity of safety configs and the difficulty of constructing high-quality demonstrations at scale" - the complexity of safety configs is a problem for this work too. Moreover, in-context learning requires only a handful of demonstrations, hence the argument of absence of demonstrations at scale is void. Hence, I think these arguments against in-context learning are not very strong.
>
> We’d like to clarify that the difficulty for in-context learning lies in the issue of constructing a set of demonstrations that fully covers the desired safety config. Given real-world safety configs as those in CoSApien, we find it very difficult to fully specify the desired safety behavior with examples alone. For example, “allowing violence but excluding descriptions of severed body parts or limbs” requires multiple carefully designed demonstrations but is easily described in natural language.
>
>
>
> > Line 183: How do you check/verify coverage? Line 405: Manually checking that the test set contains all the 3 kinds of prompts is ok, but what is ensuring that to be the case in general for all the prompts? As far as I understand, it is just a dataset containing some prompts, not necessarily according to some safety configs.
>
> We’d like to clarify that both test sets are test safety configs paired with prompts relevant to the configs (detailed in Line 173 to 183). As detailed in Appendix A.12, during the data curation stage for CoSApien, we produce three types of targeted prompts (allowed, disallowed, partial) for each test config and manually verified all prompts adhere to the desired types. For CoSAlign-Test, we conduct human verification of the automatically produced prompt risk category labels on a subset of 600 prompts and find a high human agreement rate of 89.8%. We then use this category label as proxy to select prompts and ensure each test config is covered. This is detailed in A.7 where we provide the full breakdown by prompt types.
>
> > How would GPT-4's own safety alignment affect its responses as judge-safe(.) in line 188?
>
> We kept track of the proportion of judge-safe requests that are refused by GPT-4, and find this number to be very low in all cases (less than 2% of responses). We also qualitatively find that when GPT-4 does not produce a refusal, it gives reasonably good rationales and results. Note that besides GPT-4 evaluation, we also conduct human evaluation on CoSApien and find consistent results on judge-safe.
>
> > Can the CoSA-Score be made more useful by assigning -1 to refusals/unhelpful responses?
>
> Thanks for the suggestion! We believe it’s better to give CoSA-Score to refusals/unhelpful response because these responses should be more preferred than responses that are both helpful and unsafe under the current config, which lead to negative scores. A responsible system should never give unsafe responses that are helpful to conducting harmful activities, but if the user query is not answerable given the current safety guidelines, a refusal is a reasonable response.

---

> ### Author Response · Authors · 2024-11-17
> **Response to reviewer pCoa - continued**
>
> > Line 268: Are the training prompts corresponding to or independent of any safety configs used in the training? Can the risk taxonomy made in the paper used in place of the other prior taxonomies, or is it specially tailored to CoSA?
>
> In our current pipeline, the training prompts correspond to the training safety configs because we first derive the risk taxonomy base on the training prompts (Appendix A.2), and then synthesize relevant safety configs (Line 302). But in principle, the training prompts do not need to curate together with the risk taxonomy as long as the taxonomy covers a broad range of risks relevant to the training prompts. Our derived taxonomy is not specifically tailored to CoSA and can be used in place of other prior taxonomies.
>
>
>
> > Is $C_{i, j} \in R$, in line 313?
>
> Yes, each config risk category is a subset of the taxonomy $R$.
>
>
>
> > How is judge-help conceptually different from judge-addr?
>
> While judge-help evaluates the helpfulness of a response in detail and gives a score of 0 to 5, judge-addr only considers whether the response is a refusal or not. It does not consider “how well” the model answers the response.
>
>
>
> > Line 350: Why are allowed risks penalized by $\alpha$? I think they should be rewarded, as the model is adhering to the safety config.
>
> Great question! We argue that models should only use allowed risks *as needed* in order to achieve better helpfulness. For example, if violent content is allowed in the videogame setting, the model should not be rewarded for producing violent descriptions on prompts where violence is not needed. Therefore, we give a small but positive penalty for allowed risks.
>
>
>
> > Line 363: Why is the adversarial partition of WildGuardTrain removed from the training set?
>
> We made this design choice to limit the number of training prompts due to our compute resources. Nevertheless, as shown in additional results in “Response to all reviewer” section, CoSAlign achieves increased robustness against adversarial attacks.
>
>
>
> > Line 407: Does "helpful" mean that the judge-help outputs anything > 0 or equal to 1? Similarly, what values of the helpfulness scores from humans are considered to be "helpful" in Table 4?
>
> We’d like to clarify that (detailed in Appendix A.6) for both the GPT-4 and human judges, raw scores are given on the scale of 0 to 5. A response is considered helpful if judge-help output $\geq$ 1. This ensures consistent results between GPT-4 and human judges.
>
>
>
> ## References
>
> [1] [Investigating Cultural Alignment of Large Language Models](aclanthology.org/2024.acl-long.671)
>
> [2] [Constitutional AI: Harmlessness from AI Feedback](https://arxiv.org/abs/2212.08073)
>
> [3] [Collective Constitutional AI: Aligning a Language Model with Public Input](https://arxiv.org/abs/2406.07814)
>
> [4] [A proposal for importing society’s values](https://aligned.substack.com/p/a-proposal-for-importing-societys-values)
>
> [5] [We're Afraid Language Models Aren't Modeling Ambiguity](https://arxiv.org/abs/2304.14399)

---

> > ### Comment · Reviewer_pCoa · 2024-11-22
> >
> > Thanks to the authors for their responses and new experiments. I think my score is suitable and would like to retain it. I would recommend the authors to explicitly mention in the main paper that CoSA starts off with a safety-trained model.

---

### Official Review · Reviewer_hHMw · 2024-11-04

**Soundness:** 2
**Presentation:** 3
**Contribution:** 2
**Rating:** 6
**Confidence:** 5

**Summary:**

The paper introduces CoSA, a framework designed to adapt LLMs to diverse, context-sensitive safety requirements in real-time without retraining. CoSA allows users to define safety configs, which can be adjusted on-the-fly, enabling flexible safety alignment. The framework includes CoSAlign, a method for training the model to follow these configs using synthetic preference data, and introduces CoSA-Score and CoSApien, a scoring metric and benchmark, respectively, to evaluate both helpfulness and safety adherence of model responses. CoSA demonstrates strong adaptability to unseen safety configs, promoting a pluralistic approach to safety in LLMs. However, the paper has limitations in its mathematical foundations, including formal guarantees for controllability and robustness, which could impact its theoretical reliability in diverse or adversarial scenarios.

**Strengths:**

1. CoSA enables customizable safety configurations, shifting from a one-size-fits-all model to a flexible approach, valuable for applications with varied cultural, legal, or organizational safety needs.
2. Different safety configs in real-time, allowing rapid adaptation to new safety requirements without retraining.
3. CoSAlign uses synthetic data generation, error scoring, and preference optimization to streamline fine-tuning for safety, reducing manual annotation needs for large-scale applications.

**Weaknesses:**

1. The error-scoring mechanism used in CoSA assigns arbitrary penalties for different categories of errors (small penalties for allowed risks, large for disallowed, and medium for non-responsive answers). Without a rigorous, data-driven foundation for these penalty values, there is a risk that these scores might not accurately reflect real-world preferences or safety needs.
2. Preference optimization may converge poorly due to data noise, potentially causing inconsistent or suboptimal model behavior. Convergence properties under various conditions are not well analyzed.
3. It does not establish that the model will consistently adhere to the given safety configs across different input distributions. In complex or adversarial input settings, the model might fail to control responses reliably, highlighting the need for a experimental prove or bound controllability performance.
4. CoSAlign relies heavily on synthetic preference data generated by combining safety configs with diverse prompts, but the distribution of this synthetic data may not match real-world query distributions. Mathematically, this could result in a distributional shift where the model’s learned preferences are poorly calibrated to actual use cases, limiting generalization.
5. CoSA’s preference optimization relies on a risk taxonomy with a finite set of categories. If the taxonomy is not exhaustive, the model might overfit to specific risk categories observed in training, failing to generalize to novel types of risks. This overfitting issue could be mitigated by formalizing the model’s entropy over response categories or by incorporating latent variable models to allow for broader risk representations.

**Questions:**

1.	How’s the performance on the popular metric Attack success Rate (ASR) by the framework?
2.	What is the performance on popular jailbreak methods like PAIR, DeepInception, GPTFuzzer?
3.	There was no discussion on the oversafety performances like XSTest or OKTest?
4.	I see there are plenty of papers which should come in related work or motivation of this work (training free) but they were not cited (here are some which I have found in single search but there are many more.)?

[1] SafeDecoding: Defending against Jailbreak Attacks via Safety-Aware Decoding (https://arxiv.org/abs/2402.08983),
[2] Safety Arithmetic: A Framework for Test-time Safety Alignment of Language Models by Steering Parameters and Activations
(https://arxiv.org/abs/2406.11801),
[3] SafeInfer: Context Adaptive Decoding Time Safety Alignment for Large Language Models (https://arxiv.org/abs/2406.12274),
[4] Controlled Text Generation via Language Model Arithmetic (https://arxiv.org/abs/2311.14479)

---

> ### Author Response · Authors · 2024-11-17
> **Response to reviewer hHMw**
>
> We sincerely thank reviewer `hHMw` for their thoughtful review. We are grateful that you find our controllable safety alignment method is “valuable for applications with varied cultural, legal, or organizational safety needs” and “[reduces] manual annotation needs for large-scale applications.” We hope our response below addresses your concerns:
>
> ## Response to Weaknesses
>
> **On error-scoring mechanism assign arbitrary penalties**: We’d like to clarify that the purpose of the error penalties is to ensure responses that *do not violate the safety configs* and *maintains helpfulness*, are preferred over responses that do not satisfy these two criteria. It is only used in the response pairing stage and only the relative error between two responses matters. For example, suppose there are responses A, B, and C:
>
> - Response A contain one disallowed risk but address the question -> error $\beta$
> - Response B contain one allowed risk but does not address the question -> error $\alpha+\gamma$
> - Response C contain one allowed risk and address the question -> error $\alpha$
>
> As long as $\alpha < \gamma < \beta$ is satisfied, we will prefer Response C over A, C over B, and B over A no matter what the absolute value of these hyperparameters are. We set $\alpha=0.1, \gamma=1, \beta=3$. Further tuning these hyperparameters might lead to additional gains in rare cases (e.g., when the response contains many types of allowed risks), but we empirically find that CoSAlign is already very effective (see Table 3, 4, 5 in Section 6).
>
> **On convergence properties of preference optimization**: We respectfully disagree with the reviewer that this should be listed as a weakness of our work. We do not discuss convergence properties or other math details simply because we are not proposing any preference optimization algorithm. Instead, we are just utilizing the existing DPO algorithm, one of the most commonly used and widely adopted algorithms for preference optimization, in our data-centric CoSAlign method. We clarify that our main contribution in this work (summarized already in Line 92-96) is proposing a comprehensive framework (defining the task, setup, evaluation protocol, benchmark, and the CoSAlign method) on controllable safety alignment. While the convergence properties of preference optimization are an active research area, it is not the focus of this work. Moreover, our framework makes no assumptions of the preference optimization technique used. It is straightforward to utilize more advanced algorithms such as IPO [1], KTO [2], and SimPO [3] with CoSAlign.
>
> **On input distribution shift for safety configs**: Indeed, the distribution of safety configs may change between training and real-world deployment, and maintaining sufficient controllability under config distribution shift is an important issue, and we **already conducted experiments in Section 6.1** (discussed in Line 450-464). We summarize it here again for your convenience: we conducted experiments on config distribution shift by training on CoSAlign-Train, which only contain synthetic categorical risk categories, and test on CoSApien, our proposed benchmarks consist of complex, fine-grained real-world configs (see examples in Appendix A.12). Human evaluation (Table 4) shows that CoSAlign still possesses strong controllability in this out-of-distribution setting and outperforms all baselines. This is strong evidence that CoSAlign can generalize from simpler training configs to more complex test configs. Please see Section 6.1 for more details.
>
> It is difficult to theoretically quantify the controllability of CoSAlign in all settings, where safety configs can be arbitrarily complex. Since our focus and main contributions are proposing a comprehensive framework of controllable alignment, we leave the theoretical aspect of quantifying controllability to future work.
>
> **On CoSAlign’s risk taxonomy**: We’d like to clarify that the risk taxonomy is only used during the training of CoSAlign in order to synthesize large-scale diverse and relevant preference data. Although this risk taxonomy is likely not exhaustive, we show in Table 4 that CoSAlign maintains controllability gain on the real-world CoSApien dataset and generalizes to novel risks that are more fine-grained than the training categories. Moreover, Table 3 and 5 shows CoSAlign is still effective on unseen synthetic configs (configs that contain risk categories held-out from training). These results provide plenty of evidence that our current risk taxonomy is diverse enough to not overfit to training categories. Using a risk taxonomy with a finite set of categories is also a common approach in recent works such as Llama Guard [4], BeaverTails [5], and WildGuard [6]. Therefore, we argue that risk taxonomy is an effective *practical* approach for large-scale data synthesis in controllable safety alignment.

---

> ### Author Response · Authors · 2024-11-17
> **Response to reviewer hHMw - continued**
>
> ## Response to questions
>
> **Performance on jailbreak method & ASR**: Per your suggestion, we have conducted additional experiments on 3 popular jailbreak attacks and shown the results above in the **“Response to all reviewers”** section. We find that CoSAlign lead to significantly improved adversarial robustness against these attacks.
>
>
>
> **Oversafety performance**: We conduct experiments on XSTest, shown in the **“Response to all reviewers”** section, and find CoSAlign lead to notably less refusal compared to Llama3.1-8B-Instruct. This indicates the increased controllability can allow model to better determine the safety boundary for refusal.
>
>
>
> Thank you for your suggestions of the related work on efficient alignment. Note that we already have a focused discussion on inference-time alignment which focuses on efficient training-free approaches, but we will add these to the manuscript for further completeness.
>
>
>
> ## References
>
> [1] [A General Theoretical Paradigm to Understand Learning from Human Preferences](https://arxiv.org/abs/2310.12036)
>
> [2] [KTO: Model Alignment as Prospect Theoretic Optimization](https://arxiv.org/abs/2402.01306)
>
> [3] [SimPO: Simple Preference Optimization with a Reference-Free Reward](https://arxiv.org/abs/2405.14734)
>
> [4] [Llama Guard: LLM-based Input-Output Safeguard for Human-AI Conversations](https://arxiv.org/abs/2312.06674)
>
> [5] [BeaverTails: Towards Improved Safety Alignment of LLM via a Human-Preference Dataset](https://arxiv.org/abs/2307.04657)
>
> [6] [WildGuard: Open One-Stop Moderation Tools for Safety Risks, Jailbreaks, and Refusals of LLMs](https://arxiv.org/abs/2406.18495)

---

> ### Author Response · Authors · 2024-11-23
> **Invitation for Comments and Clarifications**
>
> Dear reviewer `hHMw`,
>
> We greatly value your feedback and have provided clarifications to your questions and additional experiments on jailbreaking and over-refusal as you requested. To ensure that we have properly addressed your concerns, we would greatly appreciate if you could review our responses and provide any further comments. We look forward to engaging with you before the discussion period ends.
>
> Thank you for your time and consideration.

---

> > ### Author Response · Authors · 2024-11-25
> > **A second reminder about considering our response**
> >
> > Dear reviewer hHMw,
> >
> > Only one day is left until the end of the discussing period. Would you please consider our response and let us know if we have not addressed any of your concerns?

---

> ### Comment · Reviewer_hHMw · 2024-11-25
> **Thanks for the response**
>
> Dear Authors,
>
> Thank you for your clarifications. Based on the current state of the work, I am inclined to slightly increase my score. However, I fell that the key points I raised should have been addressed in the first version hence not able to increase much.
>
> I recommend that the authors incorporate these discussions and the referenced citations into the appendix in future revisions.
>
> Thanks.

---

> > ### Author Response · Authors · 2024-11-27
> > **Thanks for your continued engagement**
> >
> > Dear reviewer `hHMw`,
> >
> > We sincerely appreciate your follow-up response. Note that our rebuttal mainly serves to clarify potential misunderstandings in our work. We conducted additional experiments to answer your questions and address your concerns within the scope of our work.
> >
> > Regarding your follow-up message on “I fell that the key points I raised should have been addressed in the first version hence not able to increase much,” we are wondering if there are any specific key points that you feel have not yet been addressed after our rebuttal? We’d love to further improve our work based on your feedback or provide further details if needed.
> >
> > Since we have already conducted additional experiments and presented the results, we believe this should be considered equally as the results presented in the initial submission, as the ICLR discussion period is a crucial stage of the submission process and we will include all results in the camera-ready version, if accepted. Thanks so much for your continued engagement and we are always looking to improve our work with your input and if needed, provide further clarification.

---

> > > ### Author Response · Authors · 2024-12-01
> > > **A last reminder to reviewer hHMw**
> > >
> > > Dear reviewer hHMw,
> > > Tomorrow is the last day for you to respond to our rebuttal. Would you please consider our last response and let us know what you think? Thanks,

---

### Author Response · Authors · 2024-11-17
**Response to all reviewers**

We sincerely appreciate all reviewers for their detailed review. We are glad that reviewers find our proposed framework “valuable for applications with varied cultural, legal, or organizational safety needs” and are “of practical importance for effective and safe usage of LLMs in diverse applications.” Reviewers find our proposed CoSAlign method, which “properly incorporates both safety and helpfulness,” is important, and commended our human-authored CoSApien benchmark as well as the novel CoSA-Score evaluation protocol.

While we believe we have provided a comprehensive set of experiments that fully justify and support our contributions, we chose to provide additional experimental results to answer the specific questions from Reviewers `hHMw` (on jailbreak & over-refusal) and `c6ps` (on Cascade methods). While we have already conducted general safety evaluations in Section 6.2 (see Table 6 for details), and one of the benchmarks (StrongReject [1]) specifically focuses on jailbreaks, we now conduct additional experiments on jailbreak attacks and over-refusal for further soundness. Moreover, we now show the applicability of combining CoSAlign and Cascade methods. Note that because the Cascade methods are expensive, we still argue that CoSAlign (without Cascade) is the best method in practice. Please find the results below.


## Additional experiments on jailbreak attacks and over-refusal
In this section, we show that CoSAlign **significantly improves adversarial robustness against jailbreak attacks and reduces over-refusal** compared to the base model.

We use off-the-shelf Llama3.1-8B-Instruct as the baseline, and evaluate the CoSAlign-tuned variant using the safety config that indicates no type of risk is allowed. We conduct the popular GPTFuzzer [2], PAIR [3], and TAP [4] attacks on Llama3.1-8B-Instruct and our Llama3.1-8B-Instruct+CoSAlign models and report **Attack Success Rate** (lower is better) as the metric. All experiments are conducted on HarmBench [5], a standardized testing framework designed to enable fair comparisons. The results are summarized below:

| Model                         | GPTFuzzer | PAIR     | TAP      |
|-------------------------------|-----------|----------|----------|
| Llama3.1-8B-Instruct          | 68.8      | 26.3     | 32.5     |
| Llama3.1-8B-Instruct+CoSAlign | **38.8**  | **23.8** | **18.8** |

Surprisingly, we find that CoSAlign not only avoids degrading adversarial robustness but also **significantly enhances** it against popular jailbreak attacks. This result suggests that the improved safety controllability provided by CoSAlign does not conflict with adversarial robustness. We hypothesize that CoSAlign's enhanced controllability may implicitly strengthen the model’s safety reasoning capabilities, thereby making it more robust to attacks designed to “trick” LLMs into engaging in disallowed behaviors.

We also conduct experiments to investigate the over-refusal rate on XSTest [6] before and after applying CoSAlign. We report results on the safe subset, where no prompt should be refused, and present refusal rates (lower is better):

| Model                         | Full Refusal | Partial Refusal | Overall Refusal |
|-------------------------------|--------------|-----------------|-----------------|
| Llama3.1-8B-Instruct          | 8%           | 0%              | 8%              |
| Llama3.1-8B-Instruct+CoSAlign | 1.6%         | 1.2%            | **2.8%**        |

Results show that **CoSAlign significantly reduced the over-refusal rate** of Llama3.1-8B-Instruct. This indicates that the enhanced safety controllability provided by CoSAlign helps the model better reason about which prompts should be refused and which should not.

---

> ### Author Response · Authors · 2024-11-17
> **Response to all reviewers - continued**
>
> ## Additional experiments on combining CoSAlign and Cascade methods
>
> In this section, we show that **Cascade methods can be effectively incorporated into CoSAlign-tuned models** to trade off helpfulness for safety.
>
>
>
> Because the Cascade methods use a filtering model to label unsafe responses and replace them with refusals, they can also be applied on top of CoSAlign. Below, we demonstrate the effectiveness of CoSAlign+Cascade on the seen split of CoSAlign-Test (the unseen split follows the same pattern):
>
> | Model                                | CoSA-Score | Help+safe | Help+unsafe |
> |--------------------------------------|------------|-----------|-------------|
> | Llama3.1-8B-Instruct                 | 0.182      | 23.7%     | 2.0%        |
> | Llama3.1-8B-Instruct+Cascade         | 0.171      | 21.9%     | 1.6%        |
> | Llama3.1-8B-Instruct+Cascade-Oracle  | 0.201      | 23.7%     | **0.0%**    |
> | L3.1-8B-Inst-CoSAlign                | 0.408      | 52.0%     | 5.2%        |
> | L3.1-8B-Inst-CoSAlign+Cascade        | 0.368      | 45.5%     | 3.0%        |
> | L3.1-8B-Inst-CoSAlign+Cascade-Oracle | **0.454**  | **52.0%** | **0.0%**    |
>
> Results show that, similar to applying Cascade on the base Llama-3.1-8B model, applying Cascade on CoSAlign-tuned model can also reduce the rate of helpful+unsafe responses. Cascade lowers helpful+unsafe responses at the cost of decreasing helpful+safe responses, trading off helpfulness for better safety. We acknowledge that while CoSAlign+Cascade led to slightly more helpful+unsafe responses than Llama3.1-8B-Instruct+Cascade, the gap is small and CoSAlign lead to significantly improved helpful+safe responses, and thus a much higher CoSA-Score.
>
> In summary, applying Cascade-Oracle on CoSAlign achieves the highest CoSA-Score, the highest helpful+safe responses, and the lowest helpful+unsafe responses across the board, demonstrating the effectiveness of combining both methods.
>
>
>
> ## References
>
> [1] [A StrongREJECT for Empty Jailbreaks](https://arxiv.org/abs/2402.10260)
>
> [2] [GPTFUZZER: Red Teaming Large Language Models with Auto-Generated Jailbreak Prompts](https://arxiv.org/abs/2309.10253)
>
> [3] [Jailbreaking Black Box Large Language Models in Twenty Queries](https://arxiv.org/abs/2310.08419)
>
> [4] [Tree of Attacks: Jailbreaking Black-Box LLMs Automatically](https://arxiv.org/abs/2312.02119)
>
> [5] [HarmBench: A Standardized Evaluation Framework for Automated Red Teaming and Robust Refusal](https://arxiv.org/abs/2402.04249)
>
> [6] [XSTest: A Test Suite for Identifying Exaggerated Safety Behaviours in Large Language Models](https://arxiv.org/abs/2308.01263)

---

### Author Response · Authors · 2024-12-03
**Summary of the discussion period**

Dear all reviewers and chairs,



We would like to express our sincere gratitude to all reviewers for their constructive feedback and recognition of our contributions: reviewers find our framework **“valuable for applications with varied cultural, legal, or organizational safety needs”** (`hHMw`) with **“practical importance for effective and safe usage of LLMs in diverse applications”** (`pCoa`), **“more powerful and less ambiguous”** compared to existing approaches (`c6ps`), and **our proposed CoSA-Score is novel** (`6dSX`).



We have provided detailed responses and additional experiments to address all reviewer concerns:

- We have clarified to reviewer `hWMw` that convergence properties of preference optimization and theoretical quantifications of controllability is not the focus and out of scope for this work. Additionally, we have addressed their questions on error-scoring mechanism, risk taxonomy, and resolved the concern on jailbreaking and over-refusal supported by extensive experiments showing positive results.

- We have addressed concerns from reviewer `pCoa` by clarifying that CoSAlign initializes from a safety-aligned model.

- We have conducted additional experiments combining CoSAlign and Cascade methods, achieving “the best of both worlds,” and resolved reviewer `c6ps`’s concerns.

- We have also clarified reviewer `6dSX`’s concerns on core contributions and generalization abilities by referring to relevant sections and experiments in the paper.

We believe reviewers’ concerns are adequately addressed since all 4 ratings are leaning accept. Thanks again for your feedback and engagement!

---

### Meta-Review · Area_Chair_Ty2A · 2024-12-20

**Metareview:**

The paper introduces CoSA, a framework designed to adapt LLMs to diverse, context-sensitive safety requirements in real-time without retraining.

+ The topic is of interest.
+ The reviewers found the proposed CoSA framework to be interesting.

- Some of the technical details were missing or hard to follow.

**Additional Comments On Reviewer Discussion:**

There were some issues raised in the initial reviews, including
- the error scoring mechanism
- convergence properties of preference optimization
- initialization strategy
- additional experiments combining different methods
- generalization abilities.
These issues were partly addressed in the rebuttal. All the reviewers now recommend acceptance. The authors should incorporate the clarification in the rebuttal in the camera ready version.

---

### Decision · Program_Chairs · 2025-01-22

Accept (Poster)